# Clinical utility of targeted SARS-CoV-2 serology testing to aid the diagnosis and management of suspected missed, late or post-COVID-19 infection syndromes: Results from a pilot service implemented during the first pandemic wave

Nicola Sweeney[1], Blair Merrick[1,2]*, Rui Pedro Galão[3], Suzanne Pickering[3], Alina Botgros[1], Harry D. Wilson[3], Adrian W. Signell[3], Gilberto Betancor[3], Mark Kia Ik Tan[2], John Ramble[4], Neophytos Kouphou[3], Sam Acors[3], Carl Graham[3], Jeffrey Seow[3], Eithne MacMahon[1,2], Stuart J. D. Neil[3], Michael H. Malim[3], Katie Doores[3], Sam Douthwaite[1,2], Rahul Batra[1,2], Gaia Nebbia[1,2], Jonathan D. Edgeworth[1,2]

1 Department of Infectious Diseases, Guy's and St Thomas' NHS Foundation Trust, London, United Kingdom, 2 Department of Infectious Diseases, Centre for Clinical Infection and Diagnostics Research, School of Immunology and Microbial Sciences, King's College London, London, United Kingdom, 3 Department of Infectious Diseases, School of Immunology & Microbial Sciences, King's College London, London, United Kingdom, 4 Infection Sciences, Viapath LLP, St Thomas' Hospital, London, United Kingdom

* blair.merrick@nhs.net

## Abstract

During the first wave of the global COVID-19 pandemic the clinical utility and indications for SARS-CoV-2 serological testing were not clearly defined. The urgency to deploy serological assays required rapid evaluation of their performance characteristics. We undertook an internal validation of a CE marked lateral flow immunoassay (LFIA) (SureScreen Diagnostics) using serum from SARS-CoV-2 RNA positive individuals and pre-pandemic samples. This was followed by the delivery of a same-day named patient SARS-CoV-2 serology service using LFIA on vetted referrals at central London teaching hospital with clinical interpretation of result provided to the direct care team. Assay performance, source and nature of referrals, feasibility and clinical utility of the service, particularly benefit in clinical decision-making, were recorded. Sensitivity and specificity of LFIA were 96.1% and 99.3% respectively. 113 tests were performed on 108 participants during three-week pilot. 44% participants (n = 48) had detectable antibodies. Three main indications were identified for serological testing; new acute presentations potentially triggered by recent COVID-19 e.g. pulmonary embolism (n = 5), potential missed diagnoses in context of a recent COVID-19 compatible illness (n = 40), and making infection control or immunosuppression management decisions in persistently SARS-CoV-2 RNA PCR positive individuals (n = 6). We demonstrate acceptable performance characteristics, feasibility and clinical utility of using a LFIA that detects anti-spike antibodies to deliver SARS-CoV-2 serology service in adults and children. Greatest benefit was seen where there is reasonable pre-test probability and

**Data Availability Statement:** All relevant data are within the paper and its Supporting Information files.

**Funding:** King's Together Rapid COVID-19 Call awards to KJD, SJDN and RMN. MRC Discovery Award MC/PC/15068 to SJDN, KJD and MHM. National Institute for Health Research (NIHR) Biomedical Research Centre based at Guy's and St Thomas' NHS Foundation Trust and King's College London, programme of Infection and Immunity (RJ112/N027) to MHM and JE. BM was supported by an NIHR Academic Clinical Fellowship in Combined Infection Training. AWS and CG were supported by the MRC-KCL Doctoral Training Partnership in Biomedical Sciences (MR/N013700/1). GB was supported by the Wellcome Trust (106223/Z/14/Z to MHM). SA was supported by an MRC-KCL Doctoral Training Partnership in Biomedical Sciences industrial Collaborative Award in Science & Engineering (iCASE) in partnership with Orchard Therapeutics (MR/R015643/1). NK was supported by the Medical Research Council (MR/S023747/1 to MHM). SP, HDW and SJDN were supported by a Wellcome Trust Senior Fellowship (WT098049AIA). Fondation Dormeur, Vaduz for funding equipment (KJD). Development of SARS-CoV-2 reagents (RBD) was partially supported by the NIAID Centers of Excellence for Influenza Research and Surveillance (CEIRS) contract HHSN272201400008C. Viapath LLP provided support in the form of salaries for author JR, but did not have any additional role in the study design, data collection and analysis, decision to publish, or preparation of the manuscript. The specific roles of this author is articulated in the 'author contributions' section.

**Competing interests:** One of the co-authors (JR) is employed by Viapath LLP. This commercial affiliation with Viapath LLP does not alter our adherence to all PLOS ONE policies on sharing data and materials. Viapath LLP did not have any role in study design, data collection and analysis, decision to publish, or preparation of this manuscript.

results can be linked with clinical advice or intervention. Experience from this pilot can help inform practicalities and benefits of rapidly implementing new tests such as LFIAs into clinical service as the pandemic evolves.

## Introduction

Infection with SARS-CoV-2 stimulates a detectable antibody response in most people, however, the clinical utility of routine serological testing has been questioned [1, 2]. There has been uncertainty about what proportion of infected individuals produce serum antibodies, how long they persist for, what role they have in diagnosis and whether their detection provides protection against reinfection or disease manifestations upon re-exposure to the virus. These uncertainties, coupled with the fact that antibody testing for other respiratory viral infection is not standard practice and concerns regarding production and validation of rapidly developed new tests [1, 3], led to hesitancy introducing them into widespread clinical practice.

From late May 2020 the UK government prioritised serological testing to NHS staff, reserving patient testing for those interested and undergoing other blood tests with a requirement for written consent. By that time our virology department had received many enquiries from different specialties asking whether SARS-CoV-2 infection might be contributing to patient presentation despite negative or absent conventional PCR testing.

We had recently completed parallel validation of eight lateral flow immunoassay (LFIA) devices and two commercial ELISA platforms against an ELISA developed at King's College London (KCL) measuring IgG, IgA and IgM against several SARS-CoV-2 antigens (nucleocapsid (N) and spike (S) proteins and the S receptor binding domain (RBD)). Viral neutralisation assays were also established alongside the in-house ELISA to correlate antibody titres with functional activity. Validation was initially performed on a cohort of patients presenting to Guy's and St Thomas' NHS Foundation Trust and showed that the accuracy of some of the lateral flow devices was comparable to our ELISA [4].

We therefore submitted a formal request to the hospital risk & assurance board sub-committee to provide a pilot clinical SARS-CoV-2 serology service for children and adults. Pilot approval was obtained on May 29th 2020 following review of protocols and laboratory data including a further pilot validation set reported here.

## Methods

### SureScreen diagnostics LFIA validation

The CE marked SureScreen Diagnostics LFIA was selected for further validation based on results from previous head-to-head analyses [4], provision of additional proprietary information on the antigen target by the manufacturer and confidence in procurement. Tests were performed according to manufacturer's instructions by two independent operators evaluating the result as negative (0: no visible band), borderline (0.5: visible band in ideal lighting conditions, unable to photograph/ scan), positive (1: visible band in all lighting conditions), strong positive (2: visible band at the intensity of the control band or 3: visible band of greater intensity than control line) (S1 Fig). Sensitivity and specificity experiments were performed to meet MHRA validation guidance published on 19th May 2020 [5]. Serum samples were obtained from SARS-CoV-2 RNA positive (AusDiagnostics) [6] patients taken 14 or more (n = 301) and 20 or more (n = 204) days post onset of symptoms (POS) and 300 pre-pandemic samples. This included 200 stored serum samples and a panel of 100 stored acute and convalescent

confounder samples taken from individuals with EBV, CMV, HIV and a range of other viral, bacterial and fungal pathogens. 95% confidence intervals were determined using the Wilson/ Brown Binomial test. Sera from individuals diagnosed with seasonal coronaviruses were not available for testing. The research reagent for anti-SARS-CoV-2 Ab (NIBSC 20/130) obtained from the National Institute for Biological Standards and Control (NIBSC), UK, was used as a positive control for reproducibility and limit of detection experiments (IgG only) [7].

## Service delivery

Internal governance approval for service delivery was obtained based on the laboratory validation data, clinical oversight, confirmation of an ability to request and report tests on electronic systems, a review of risks and their mitigation and agreement to report back on completion of the pilot. Service commenced on 29[th] May 2020 and was delivered by scientists from the KCL Department of Infectious Diseases who had conducted all the LFIA validations. Tests were performed in and provided by the Guy's and St Thomas' Hospital Centre for Clinical Infection and Diagnostics Research (CIDR), located adjacent to hospital routine diagnostic virology and blood sciences laboratories on the St Thomas' Hospital site.

Availability of SARS-CoV-2 serology service was communicated through clinical networks with requests vetted by the clinical virology team. Samples were requested as part of routine laboratory testing route and serology was performed once daily, Monday to Friday. A positive band for either IgM or IgG (or borderline band in both) was reported to the clinician as "antibodies detected". Results were uploaded onto hospital electronic patient records as a scanned image of the lateral flow cassette with a written comment alongside telephoning where appropriate. Differential detection of IgM and IgG was not considered as part of verbal or written advice. Repeat testing was recommended when there was a high index of clinical suspicion and no antibodies were detected, or a borderline result in IgM or IgG was the only observed band (S2 Fig). A standard set of demographics, clinical information, and SARS-CoV-2 PCR results were recorded for each participant and stored in a clinical database (S3 Fig). Informal verbal or written feedback from clinicians about their views on utility was also recorded.

## ELISA

ELISA testing was performed on the 168 stored samples where sufficient sample was available from patients that had all severities of COVID-19 for comparison with the LFIA validation cohort (n = 301). All serum samples from the pilot service (where sufficient sample was available) were also batched for comparative testing at a time remote from clinical decision making.

High-binding ELISA plates (Corning, 3690) were coated with antigen (N, S) at 3 µg/mL (25 µL per well) in PBS. Wells were washed with PBS-T (PBS with 0.05% Tween-20) and then blocked with 100 µL 5% milk in PBS-T for 1 hr at room temperature. Wells were emptied and sera diluted at 1:50 in milk was added and incubated for 2 hr at room temperature. Control reagents included CR3009 (2 µg/mL), CR3022 (0.2 µg/mL), negative control plasma (1:25 dilution), positive control plasma (1:50) and blank wells. Wells were washed with PBS-T. Secondary antibody was added and incubated for 1 hr at room temperature. IgM was detected using goat-anti-human-IgM-HRP (1:1,000) (Sigma: A6907), IgG was detected using goat-anti-human-Fc-AP (1:1,000) (Jackson: 109-055-043-JIR). Wells were washed with PBS-T and either Alkaline Phosphatase (AP) substrate (Sigma) was added and read at 405 nm (AP) or 1-step TMB substrate (Thermo Scientific) was added and quenched with 0.5 M $H_2SO_4$ before reading at 450 nm (HRP). Antibodies were considered detected if OD values were 4-fold or greater above background.

### Neutralising antibody assay

Neutralising antibody testing was performed on six patients (all in infection control/ immuno-suppression management group–Fig 3). Neutralisation were conducted as previously described [8]. Serial dilutions of serum samples were prepared with DMEM media and incubated with pseudotyped HIV virus incorporating the SARS-CoV-2 spike protein [9] for 1-hour at 37°C in 96-well plates. Next, HeLa cells stably expressing the ACE2 receptor (provided by Dr James Voss, The Scripps Research Institute) were added and the plates were left for 72 hours. Infection level was assessed in lysed cells with the Bright-Glo luciferase kit (Promega), using a Victor™ X3 multilabel reader (Perkin Elmer). $ID_{50}$ for each serum was calculated using GraphPad Prism. Neutralisation titres were classified as low (50–200), moderate (201–500), high (501–2000), or potent (2001+).

### Patient and public involvement

Patients were not involved in the development of the study or its outcome measures, conduct of the research, or preparation of the manuscript.

### Ethical approval

All work was performed in accordance with the UK Policy Framework for Health and Social Care Research, and approved by the Risk and Assurance Committee at Guy's and St Thomas' NHS Foundation Trust. Informed consents were not required from participants in this study as per the guidelines set out in the UK Policy Framework for Health and Social Care Research and by the registration with, and express consent of the host institution's review board.

## Results

LFIA validation was performed using serum samples from 301 PCR-confirmed SARS-CoV-2 positive individuals collected 14 or more days POS and 300 pre-pandemic serum samples including 100 (acute and convalescent) from patients with a range of other infections that could give rise to a false positive result (Fig 2A) and in accordance with published MHRA guidance at the time. A random selection of 168 (of the 301) samples from patients with a range of disease severities (and where sufficient sample for analysis was available) were compared head-to-head with an in-house ELISA for IgM and IgG to N, S and RBD (Fig 1). Sensitivity at 14 and 20 days or more POS was 94.4% and 96.1% respectively and specificity was 99.3% (Fig 2B). Limit of detection based on visual inspection of LFIA bands by two operators was determined using the NIBSC reference standard to a dilution of 1 in 500. This was consistent with the expected limit of detection of the NIBSC in-house assay (S4 Fig) [9].

The pilot service was commenced on May 29[th] 2020 and lasted 3 weeks. 48/108 (44%) participants had detectable IgG and/or IgM SARS-CoV-2 antibodies on their first serum sample that was communicated to referring clinicians as "antibodies detected" (Fig 3). 38/48 (79%) had IgM and 47/48 (98%) had IgG bands. Five participants with a high index of suspicion but no detectable antibodies had a further serum sample tested at least one week after initial testing. All repeat samples had no detectable antibodies. Rationale for testing broadly fell into three referral categories. First, acute presentations with new symptoms potentially triggered by SARS-CoV-2 infection. This included suspected cases of Paediatric Inflammatory Multisystem Syndrome Temporally associated with SARS-CoV-2 (PIMS-TS) (n = 30), plus adults (n = 27) and children (n = 5) presenting with other clinical syndromes including thrombotic events such as strokes and pulmonary emboli (collectively called COVID-19 syndromes). Second, suspected "missed" diagnoses in individuals with a (recent) COVID-19 compatible illness who

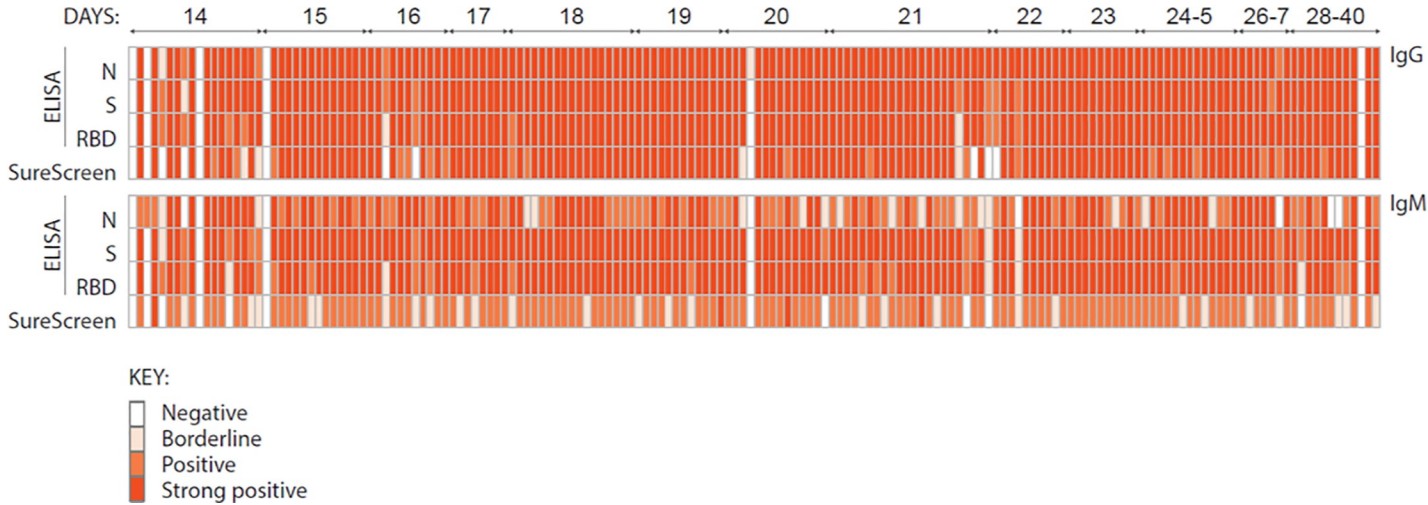

**Fig 1. Comparative assessment of 168 serum samples from SARS-CoV-2-infected individuals by ELISA and lateral flow immunoassay.** 168 serum samples from individuals with confirmed SARS-CoV-2 infection were tested for the presence of antibody by ELISA to the full spike (S), receptor binding domain (RBD) and nucleocapsid (N), and by SureScreen lateral flow immunoassay. Detection of IgG is shown in the top panel, and IgM in the bottom panel. Samples are arranged according to days post onset of symptoms, ranging from 14 to 40 days. Results are displayed as a heatmap, with white indicating a negative result, and gradations of orange indicating the magnitude of response detected.

either never had an RNA test performed (n = 19) or viral RNA was not detected in respiratory specimens (n = 21). Third, those for whom antibody detection made a significant contribution to decisions on infection control management or immunosuppressive treatment (n = 6).

Of 30 children with suspected PIMS-TS, 11 had detectable antibodies (37%). Reviewing the clinical history of the 19 with no detectable antibodies, seven had an alternate plausible diagnosis, or did not fulfil PIMS-TS diagnostic criteria at the time of discharge and 12 had ongoing high clinical suspicion of PIMS-TS. Two children (participants 018 and 034) had repeat testing at least seven days later, neither had detectable antibodies at this stage. For the remaining 32 PCR negative/ not done individuals presenting with a potential post-COVID syndrome, seven (21.2%) had antibodies detected. This included two with the diagnosis of pulmonary embolism (PE), one with a new diagnosis of interstitial lung disease (ILD), two with a hyperinflammatory syndrome (akin to PIMS-TS), one patient with a relapse of HSV encephalitis, and one patient with paracentral acute middle maculopathy.

40 individuals were tested to identify potential missed COVID-19 diagnoses comprising nine presenting to hospital with ongoing compatible symptoms but negative SARS-CoV-2 RNA tests, and 31 who had recovered from a recent compatible illness in the community, including 15 individuals with end-stage renal failure, who had been advised to shield, and 12 patients attending the respiratory led post-COVID clinic due to failure to return to their baseline level of function. Overall, 24/40 (60%) had detectable antibodies, including two patients admitted to ITU but with repeatedly negative PCR results on upper and lower respiratory sampling.

Of the six individuals with persistent SARS-CoV-2 RNA on nose and throat swabs tested to guide infection control or immunosuppression decisions, all had detectable antibodies on SureScreen LFIA, and when tested, moderate (n = 1, ID50 = 277), high (n = 1, ID50 = 1135), or potent (n = 4, ID50 = 2333, 4130, 5164, 5248) neutralising antibodies titres. This implied, when considered with other factors such as time from first positive PCR test, and threshold cycle for RNA detection, that they were no longer infectious, and had a degree of protection from reinfection.

| Panel | Samples tested (n) | Positive (n) | Specificity (95% CI) |
|---|---|---|---|
| Pre-pandemic (from March 2019) | 200 | 1* | 99.5 % |
| Cytomegalovirus (CMV) | 8 | 0 | 100 % |
| Epstein-Barr virus (EBV) | 10 | 0 | 100 % |
| Hepatitis A virus | 8 | 0 | 100 % |
| Hepatitis B virus | 7 | 0 | 100 % |
| Hepatitis C virus | 5 | 0 | 100 % |
| Human immunodeficiency virus (HIV) | 9 | 0 | 100 % |
| Kaposi's sarcoma herpesvirus ½ | 5 | 0 | 100 % |
| Measles virus | 6 | 1* | 83.3 % |
| Mumps | 9 | 0 | 100 % |
| Mycobacterium | 1 | 0 | 100 % |
| Parvovirus | 7 | 0 | 100 % |
| Pneumocystis pneumonia | 4 | 0 | 100 % |
| Rubella virus | 5 | 0 | 100 % |
| Syphilis virus | 4 | 0 | 100 % |
| Toxoplasma gondii | 7 | 0 | 100 % |
| Varicella zoster virus | 5 | 0 | 100 % |
| Confounder samples (all) | 100 | 1 | 99.0 % |
| Overall | 300 | 2 | 99.3% (97.6-99.8) |

*False positives were borderline bands in either IgM or IgG. A similar result in the pilot triggered a request for a repeat sample to test.

| Panel | Samples tested (n) | Positive (n) | Sensitivity (95% CI) |
|---|---|---|---|
| SARS-CoV-2-positive 14+ days POS | 301 | 284 | 94.4 % (91.1-96.4) |
| SARS-CoV-2-positive 20+ days POS | 204 | 196 | 96.1 % (92.4-98.0) |

POS= post onset of symptoms

**Fig 2. a:** Testing of samples that were pre-pandemic from patients with other infectious diseases and known confounders to estimate specificity of the SureScreen lateral flow immunoassay. **b:** Sensitivity estimates of SureScreen lateral flow immunoassay using serum samples obtained from SARS-CoV-2 PCR positive patients at greater than 14 and 20 days post reported onset of symptoms. POS = post onset of symptoms.

Serological testing was performed no earlier than 21 days post onset of symptoms (POS), up to approximately 90 days POS (where symptom onset data was available) for all participants.

When considering the combined (IgM and IgG) anti-spike ELISA data versus the SureScreen LFIA result there were 13 discrepant results. Reviewing the ELISA IgG anti-spike data only, there was greater concordance (four discrepant). The ELISA did not detect antibodies in two cases–participants 055 and 098 (where the SureScreen LFIA did), and in two cases antibodies were detected by the ELISA–participants 019 and 046 (where the SureScreen LFIA detected none). There was one individual (participant 045) with detectable anti-nucleocapsid IgG by ELISA who did not have anti-spike IgG (S3 Fig). The SureScreen LFIA did not detect antibodies in this participant–an expected result as the device only detects anti-spike antibodies. The IgM anti-nucleocapsid ELISA data has not been considered as previous work recognised the low specificity of this test (4). ELISA testing was not performed on three participants due to lack of sample availability (participants 002, 036 and 096).

## Discussion

This pilot SARS-CoV-2 serology service was introduced two months after the peak of acute UK COVID-19 admissions and provided results on 108 patients over a three-week period. It included a large number of children presenting with a new hyper-inflammatory, Kawasaki-like syndrome, termed PIMS-TS [10], to the on-site Evelina London Children's Hospital that provides regional specialist services. 37% had antibodies detected, lower than previously reported [10, 11], potentially due to increased awareness and broadening of clinical evaluation criteria, supported by a number of children having this diagnosis removed from discharge coding.

Serology was particularly helpful aiding diagnosis and management of what is an increasing range of assumed COVID-19 triggered conditions [12–17]. For example, antibodies were detected in two patients presenting with a PE that was therefore considered a provoked event, limiting the need for additional investigations and reducing the period of anticoagulation. Negative serology also helped to discount, although recognising the limitations of testing, could not fully exclude COVID-19 as a potential trigger for newly presenting conditions. These included acquired haemophilia A and a range of unusual dermatological presentations e.g. 'Covid toes'.

Detecting antibodies in patients with persistently positive SARS-CoV-2 PCR tests despite symptom resolution, a phenomenon reported elsewhere [18], enabled important decisions for infection control and immunosuppression. These decisions were supported by data that antibodies against spike protein (personal communication with SureScreen Diagnostics Ltd) correlate with neutralisation [19] and there is published guidance that neutralisation can be used as a proxy for reduced risk of transmission [20, 21]. Since neutralising experiments are time-consuming and complex, rapid tests that detect antibodies against spike, such as the SureScreen LFIA and some, but not other technologies [22–24] are a practical alternative [25] when considered alongside other factors including timing from symptom onset, ongoing symptoms, and cycle threshold or take-off values of PCR results.

| Category | Direct care team | | IP | OP | RNA result | | | Antibody result | |
|---|---|---|---|---|---|---|---|---|---|
| | | | | | + | - | ND | + | - |
| Presentation associated with SARS-CoV-2 (n=62) | Medicine | | | | | | | | |
| | - Acute | Cardiology | 5 | - | - | 5 | - | 1 | 4 |
| | - Dermatology | Haematology | 2 | - | - | 2 | - | - | 2 |
| | - ID/HIV | Intensive care | - | 4 | - | - | 4 | - | 4 |
| | - Nephrology | Respiratory | 3 | - | - | 3 | - | - | 3 |
| | | | 2 | - | - | 2 | - | 1 | 1 |
| | | | 2 | - | - | 1 | 1 | 2 | - |
| | | | 1 | - | - | 1 | - | - | 1 |
| | | | 5 | 1 | - | 5 | 1 | 1 | 5 |
| | Paediatrics | | | | | | | | |
| | - PIMS-TS | | 30 | - | 1 | 26 | 3 | 11 | 19 |
| | - Other | | 5 | - | - | 5 | - | 1 | 4 |
| | Surgery | | | | | | | | |
| | - Ophthalmology | | - | 1 | - | - | 1 | 1 | - |
| | - Urology | | 1 | - | - | 1 | - | - | 1 |
| Suspected 'missed' diagnosis of COVID-19 (n=40) | GP | | - | 2 | - | - | 2 | 2 | - |
| | Medicine | | | | | | | | |
| | - Acute | ID/HIV | 4 | - | - | 4 | - | 1 | 3 |
| | - Intensive care | Nephrology | 1 | 1 | - | 2 | - | 1 | 1 |
| | - Obstetrics | Oncology | 3 | - | - | 3 | - | 2 | 1 |
| | - Respiratory | | - | 15 | - | 5 | 10 | 11 | 4 |
| | | | 1 | - | - | - | 1 | 1 | - |
| | | | - | 1 | - | - | 1 | - | 1 |
| | | | - | 12 | - | 7 | 5 | 6 | 6 |
| Infection control/ immunosuppression management (n=6) | Medicine | | | | | | | | |
| | - Nephrology | | - | 2 | 2 | - | - | 2 | - |
| | - Oncology | | 1 | 3 | 4 | - | - | 4 | - |
| Total (n=108) | | | 66 | 42 | 7 | 72 | 29 | 48 | 60 |

IP = inpatient, OP = outpatient, ND = not done, ID = infectious diseases, PIMS-TS = paediatric inflammatory multisystem syndrome temporally associated with SARS-CoV-2, HIV = human immunodeficiency virus

**Fig 3. Referral characteristics and RNA results of individuals having SARS-CoV-2 serology testing performed during the pilot.**

The strength of this study includes the extensive prior comparison of multiple technologies using a large panel of serum samples to inform choice and validation of the selected LFIA for clinical service. Results were also consistent with recommendations from a Cochrane review published after completion of our pilot, which suggested a benefit for serology to confirm a COVID-19 diagnosis in patients who did not have SARS-CoV-2 RNA testing performed, or who had a negative result despite an ongoing high index of clinical suspicion [3].

It was also offered across the hospital to assess the broad potential clinical utility. With high pre-test probability (e.g. 45%), the positive predictive value (PPV) is 99.2%, with an acceptable negative predictive value (NPV) of 96.9%. However, it is of note that if testing were to be extended to a population where prevalence is low (e.g. 5%) the PPV falls to <90%. This re-enforces the importance of providing serology for defined patient cohorts where the pre-test probability is high and the potential clinical utility is understood [26, 27].

The main limitation of this study is in being performed at a single-centre at a discrete time-point in the COVID-19 pandemic. Since that time there have been many changes in the epidemiology and approach to COVID-19 testing. Most countries are in the midst of a second wave and vaccination will change the utility and interpretation of antibody detection. PCR tests are also more widely available to patients in the community (pillar 2) and hospital laboratories have higher capacity and more rapid PCR tests (pillar 1), which could reduce the number of missed or delayed diagnoses. There are also more accurate laboratory serology technologies, including the ability to assess dynamic responses [28–30] alongside T cell assays [31, 32], which could reduce utility of LFIA in many settings.

The discrepant ELISA and LFIA data illustrate the challenges of any single technology employed to detect specific antibodies induced in response to infection rather than cross-reactivity or anamnestic responses–particularly for IgM. Ten participants with no antibodies detected using SureScreen LFIA had low level anti-spike IgM antibodies detected by ELISA (but not IgG). At 21 or more days POS the one would expect the vast majority of individuals to have seroconverted to IgG (only one study participant had IgM only identified on SureScreen LFIA). When taken into consideration with previous validation work using this ELISA [4], these results could merely represent non-specific reactivity. The explanation for the four SureScreen results that were discrepant with the ELISA anti-spike IgG would require further investigation including repeat sampling and testing using other technologies. All LFIA results in this pilot were communicated in the context of the clinical history and the decisions being made, and where limitations of serological results in general and these technologies in particular were understood. Technologies for confirmatory testing alongside participation in external quality assurance schemes would be required to extend delivery of such a service.

Nevertheless, LFIAs are quick (10-minute test), inexpensive and are used in diagnostic laboratories for example in detecting pneumococcal and legionella urinary antigens. It is hard to predict where future clinical need for serology LFIAs might be. They could be developed further for deployment in settings with limited laboratory facilities, enabled by methods to collect capillary blood (although this will require further validation work), or used longitudinally to assess waning response to mass population vaccination campaigns where rapid high-volume longitudinal assessments might be required. There are also now technologies available for electronic reading of bands that would take away the subjectivity of reading band signal-strength by eye, which we recorded here in a semi-quantitative way by two independent observers. This experience may help inform approach to reading and communicating SARS-CoV-2 antigen lateral flow devices that are now being used by healthcare staff, patients and public [33], although the significance of band strength for both antibody and antigen lateral flow assays still requires further investigation.

This study also represents the final phase of a translational research pathway completed in three months from the basic science, comparative evaluation and now this pilot service study. The diagnostic response to this pandemic will come under continued scrutiny and there are lessons to be learnt on the ability of diagnostic laboratories and translational research teams to work together in the response to rapidly emerging infections [34]. The strengths and limitations of conducting this study at this time therefore also provides useful data to inform discussion on the requirements of academia to respond in a pandemic setting.

## Supporting information

**S1 Fig. Scanned images of LFIA cassettes for participants 084 (left) and 086 (right) labelled with band intensities.** Negative = 0: No visible borderline = 0.5: A visible band in ideal lighting conditions, positive = 1: A visible band in all lighting conditions, strong positive = 2: A visible band at the intensity of the control line or 3: A visible band of greater intensity than the control line. NB: Bands of 0.5 intensity are unable to be scanned/ photographed and therefore appear blank on the scanned image below.
(DOCX)

**S2 Fig. Flowchart of service delivery, same-day service, Monday to Friday.**
(DOCX)

**S3 Fig. Cohort demographics including age, sex, category, direct care team, RNA result (if performed), SureScreen LFIA results (band intensity recorded from 0.5–3) and ELISA data (results expressed as fold change above background, ≥4 fold above background in either IgM or IgG is reported as positive).**
(DOCX)

**S4 Fig. IgG limit of detection for the SureScreen LFIA.** Defined dilutions of the NIBSC research reference reagent for anti-SARS-CoV-2 antibody (20/130) were tested in triplicate on the SureScreen LFIA. Results are displayed as a heat map, with white indicating a negative result and gradations of orange representing the magnitude of response detected.
(DOCX)

## Acknowledgments

We are extremely grateful to all staff in Viapath Infection Sciences and Department of Infectious Diseases based at St Thomas' Hospital who helped deliver this service.

## Author Contributions

**Conceptualization:** Nicola Sweeney, Blair Merrick, Rui Pedro Galão, Suzanne Pickering, Alina Botgros, Harry D. Wilson, Adrian W. Signell, Gilberto Betancor, Mark Kia Ik Tan, John Ramble, Eithne MacMahon, Stuart J. D. Neil, Michael H. Malim, Katie Doores, Sam Douthwaite, Rahul Batra, Gaia Nebbia, Jonathan D. Edgeworth.

**Data curation:** Nicola Sweeney, Blair Merrick, Rui Pedro Galão, Suzanne Pickering, Alina Botgros, Harry D. Wilson, Adrian W. Signell, Gilberto Betancor, Mark Kia Ik Tan, Neophytos Kouphou, Sam Acors, Carl Graham, Jeffrey Seow, Katie Doores.

**Formal analysis:** Nicola Sweeney, Blair Merrick, Rui Pedro Galão, Suzanne Pickering, Harry D. Wilson, Adrian W. Signell, Gilberto Betancor, Mark Kia Ik Tan, Neophytos Kouphou, Sam Acors, Carl Graham, Jeffrey Seow, Katie Doores.

**Investigation:** Nicola Sweeney, Blair Merrick, Rui Pedro Galão, Suzanne Pickering, Alina Botgros, Harry D. Wilson, Adrian W. Signell, Gilberto Betancor, Mark Kia Ik Tan, John Ramble, Neophytos Kouphou, Sam Acors, Carl Graham, Jeffrey Seow, Katie Doores.

**Methodology:** Nicola Sweeney, Blair Merrick, Rui Pedro Galão, Suzanne Pickering, Alina Botgros, Harry D. Wilson, Adrian W. Signell, Gilberto Betancor, Mark Kia Ik Tan, John Ramble, Eithne MacMahon, Katie Doores, Sam Douthwaite, Rahul Batra, Gaia Nebbia.

**Project administration:** Nicola Sweeney, Blair Merrick, Rui Pedro Galão, Suzanne Pickering, Alina Botgros, Harry D. Wilson, Adrian W. Signell, Gilberto Betancor, Mark Kia Ik Tan, Sam Douthwaite, Rahul Batra, Gaia Nebbia, Jonathan D. Edgeworth.

**Resources:** Blair Merrick, Suzanne Pickering, John Ramble, Jonathan D. Edgeworth.

**Supervision:** Alina Botgros, Eithne MacMahon, Stuart J. D. Neil, Michael H. Malim, Katie Doores, Sam Douthwaite, Rahul Batra, Gaia Nebbia, Jonathan D. Edgeworth.

**Validation:** Nicola Sweeney, Rui Pedro Galão, Suzanne Pickering, Alina Botgros, Harry D. Wilson, Adrian W. Signell, Gilberto Betancor, Mark Kia Ik Tan, John Ramble, Neophytos Kouphou, Sam Acors, Carl Graham, Jeffrey Seow, Katie Doores.

**Writing – original draft:** Nicola Sweeney, Blair Merrick, Rui Pedro Galão, Suzanne Pickering, Alina Botgros, Katie Doores, Jonathan D. Edgeworth.

**Writing – review & editing:** Nicola Sweeney, Blair Merrick, Rui Pedro Galão, Suzanne Pickering, Alina Botgros, Eithne MacMahon, Stuart J. D. Neil, Michael H. Malim, Katie Doores, Sam Douthwaite, Rahul Batra, Gaia Nebbia, Jonathan D. Edgeworth.

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
