## [Decision Letter · Decision Letter 0]

20 Nov 2020

PONE-D-20-30782

Clinical utility of targeted SARS-CoV-2 serology testing to aid the diagnosis and management of suspected missed, late or post-COVID-19 infection syndromes: results from a pilot service

PLOS ONE

Dear Dr. Merrick,

Thank you for submitting your manuscript to PLOS ONE. After careful consideration, we feel that it has merit but does not fully meet PLOS ONE’s publication criteria as it currently stands. Therefore, we invite you to submit a revised version of the manuscript that addresses the points raised during the review process.

One issue must be clarified to meet the publication requirements of PLOS ONE: are these data new or did the authors already report them in a previous publication? Besides, the conclusions are not fully supported by the data. We invite the authors to carefully address the comments of the reviewers.

We look forward to receiving your revised manuscript.

Kind regards,

Michael Nagler, M.D., Ph.D., MSc

Academic Editor

PLOS ONE

Journal Requirements:

2. Please include your tables as part of your main manuscript and remove the individual files. Please note that supplementary tables (should remain/ be uploaded) as separate "supporting information" files

3.Thank you for stating the following in the Financial Disclosure section:

[King’s Together Rapid COVID-19 Call awards to KJD, SJDN and RMN.

MRC Discovery Award MC/PC/15068 to SJDN, KJD and MHM.

National Institute for Health Research (NIHR) Biomedical Research Centre based at Guy's and St Thomas' NHS Foundation Trust and King's College London, programme of Infection and Immunity (RJ112/N027) to MHM and JE.

AWS and CG were supported by the MRC-KCL Doctoral Training Partnership in Biomedical Sciences (MR/N013700/1).

GB was supported by the Wellcome Trust (106223/Z/14/Z to MHM).

SA was supported by an MRC-KCL Doctoral Training Partnership in Biomedical Sciences industrial Collaborative Award in Science & Engineering (iCASE) in partnership with Orchard Therapeutics (MR/R015643/1).

NK was supported by the Medical Research Council (MR/S023747/1 to MHM).

SP, HDW and SJDN were supported by a Wellcome Trust Senior Fellowship (WT098049AIA).

Fondation Dormeur, Vaduz for funding equipment (KJD).

Development of SARS-CoV-2 reagents (RBD) was partially supported by the NIAID Centers of Excellence for Influenza Research and Surveillance (CEIRS) contract HHSN272201400008C.

No sponsors or funders played a role in the study design, data collection or analysis, decision to publish, or preparation of the manuscript.].   

We note that one or more of the authors are employed by a commercial company: Viapath Group LLP

Reviewers' comments:

Reviewer's Responses to Questions

**Comments to the Author**

1. Is the manuscript technically sound, and do the data support the conclusions?

Reviewer #1: Partly

Reviewer #2: Partly

2. Has the statistical analysis been performed appropriately and rigorously? 

Reviewer #1: Yes

Reviewer #2: I Don't Know

3. Have the authors made all data underlying the findings in their manuscript fully available?

Reviewer #1: No

Reviewer #2: Yes

4. Is the manuscript presented in an intelligible fashion and written in standard English?

Reviewer #1: Yes

Reviewer #2: Yes

5. Review Comments to the Author

Reviewer #1: The submitted manuscript of Sweeney et al. highlights the potential clinical benefit of targeted SARS-CoV-2 serology testing/service to aid the diagnosis and management of suspected missed, late or post-COVID-19 infection syndrome based on a self-validated LFIA test. In general, people are strongly engouraged to design studies that investigate performance and application of serological tests for COVID-19 based on different reasons. Given the tremendous improvements in state of the art serology tests (ELISAs, flow cytometry-based assay for SARS-CoV-2-binding antibodies, Antigen tests) in the last couple of months the rapidly processed LFIA test lose its advantage over accuracy, quantification and speed compared to other tests. These limitations have also been discussed in the manuscript. Therefore, rapidly processed and accurate serology can be performed with more sophisticated tests than the LFIA in short time. Clinical advice or interventions and management based on the LFIA data summarized in this study is intended just for a defined patient cohort and needs further investigations on solid and statistically unbiased data sets to be competitive and attractive over other tests. The following major and minor concerns have to be addressed in order to warrant publication of the study.

Main concerns:

1 In a previously published study (Plos pathogen (2020) Pickering et al.) the authors already extensively validated/compared LFIA devices including the commercially available SureScreen LFIA. Therefore, the comprehensive LFIA validation shown in figure 1 re-produces their own initially published data but is not new. This also raises the question of whether another patient cohort has been used for the analysis shown in this study. Additionally, raw data points and group clustering strategy are missing.

2 The sensitivity of the LFIA test in table 1b has also already been described in the earlier study (Plos pathogen, Pickering et al. 2020, Figure 5a).

3 The high sensitivity (accuracy) of the test might derive from the biased sample group which is 14 and 20 days POS (Peak of AB response) and would rather not reflect the “real-life” situation in clinics. Therefore, clinical interpretation of results might rather be difficult (no quantification of the AB response) if random samples should/will be tested and interpreted. This is further highlighted in the study of Pickering et al. (Plos pathogen (2020) where the sensitivity (of all tests, including SureScreen) is highly dependent on the “days after onset of symptoms”.

4 Quantification and interpretation of antibody dynamics is not possible with this assay. Furthermore, additional data on IgA levels in nasopharyngeal swaps would be helpful and informative and would strengthen the considered serology service in clinics.

5 In addition to the LFIA limitations described above, serology only identifies a proportion of recently infected patients within a short time frame. Thus, seronegative results (more than 60% of the cases, table 2) are difficult to interpret and resulting interventions as well as management would be extremely difficult and speculative with this test. CD4+/CD8+ T cell memory assays might close the gap (Peng et al.2020, Nature immunology).

6 Split of patient samples into different pathologies (Table 1a) is fairly interesting but loses its statistic power (small sample size).

7 “Neutralising antibody titers” are missing from the neutralising antibody assay described in methods and results.

8 Within this small cohort, rational for testing and the resulting interpretation is highly speculative and demonstrate rather a case report.

Minor concerns:

1 Study design (flow chart) would be helpful to follow

2 Age characteristics of individual patients would be helpful

3 The SureScreen LFIA was selected based on which criteria?

4 How did you exactly correlate/categorize LFIA results with ELISA data into the different categories as “negative” ,“borderline”, “positive” and “strong positive”?

5 Neutralising antibody titers for 6 individuals with persistent SARS-CoV-2 are missing

6 Can you run blood samples on the LFIA and do those results correlate with serum samples (much faster processed) on the LFIA? Which dilution is required (serum/blood)?

7 Rapid decline in AB titers of SARS-CoV-2 infected patients is controversially discussed and should be interpreted carefully.

Reviewer #2: In this manuscript, the authors describe an extended validation of a LFIA for the rapid serological assessment of anti-SARS-CoV-2 antibodies, as well as a pilot study using this test in the clinical setting to decide whether PCR-negative patients with Covid-19 like symptoms may be infected.

In my opinion, this manuscript may be accepted with major revisions.

Here, I will raise some questions, which should be addressed in the revised manuscript

1. In the Methods and Results section, the authors mention the assessment of Limit of Detection, but there are no Results shown. Visualization of these experiments would help to interpret the heat-map results shown in figure 1. Are these LoD dilutions mentioned in the results for IgM or IgG?

2. How are the 168 samples in fig. 1 selected?

3. Table 1b: are the values for sensitivity and specificity for IgG, IgM or a combination of both?

4. Are borderline results calculated as positive or negative?

5. It is difficult to understand the sensitivity and specificity values, as 5 of 19 sera at d14 are negative. Are these samples representative?

6. I propose that the results of the pilot study are presented in the same manner as the results in Figure 1, i.e. strength of reaction for IgG and IgM, alongside with the time point after POS.

7. Paragraph 3 of results: the indications of % in brackets are misleading, as the 29 pediatric patients with PIMS-TS are 100%, the following numbers should be adapted accordingly.

8. In the same paragraph are 7 patients mentioned, but only 6 described – what about the last patient?

9. Paragraph 5 of results: the implication drawn by the authors is not very robust. There is no evidence for this implication, as results are lacking. Is there correlation between the threshold cycle of RNA detection and the strength of Antibody signal in the LFIA? Furthermore, the results of the neutralization experiments should be included in the comparison of RT-PCR and LFIA results for these follow-up patients.

10. The authors claim, that this pilot should address the question, whether the serological rapid antibody test may help in the clinical routine in decisions. as mentioned in the methods, the clinicians using the service described, have been contacted for informal feedback and their view of utility. These feedbacks are not systematically evaluated.

11. Paragraph 5 of discussion: the first sentence does not help a lot.

12. Paragraph 5 of discussion: how are the inclusion criteria set, to obtain high pre-test probability? Could you explain how the PPV and NPV with the different pre-test probabilities are calculated?

6. PLOS authors have the option to publish the peer review history of their article (what does this mean?). If published, this will include your full peer review and any attached files.

Reviewer #1: No

Reviewer #2: No

---

## [Author Response · Author response to Decision Letter 0]

21 Dec 2020

We have uploaded a response to the Reviewers as a separate file entitled 'Response to Reviewers' as requested by the Editor.

---

## [Decision Letter · Decision Letter 1]

14 Jan 2021

PONE-D-20-30782R1

Clinical utility of targeted SARS-CoV-2 serology testing to aid the diagnosis and management of suspected missed, late or post-COVID-19 infection syndromes: results from a pilot service

PLOS ONE

Dear Dr. Merrick,

Thank you for submitting your manuscript to PLOS ONE. After careful consideration, we feel that it has merit but does not fully meet PLOS ONE’s publication criteria as it currently stands. Therefore, we invite you to submit a revised version of the manuscript that addresses the points raised during the review process.

The reviewer still raise major concerns and I would like to encourage the authors to address these issues.

We look forward to receiving your revised manuscript.

Kind regards,

Michael Nagler, M.D., Ph.D., MSc

Academic Editor

PLOS ONE

Reviewers' comments:

Reviewer's Responses to Questions

**Comments to the Author**

1. If the authors have adequately addressed your comments raised in a previous round of review and you feel that this manuscript is now acceptable for publication, you may indicate that here to bypass the “Comments to the Author” section, enter your conflict of interest statement in the “Confidential to Editor” section, and submit your "Accept" recommendation.

Reviewer #1: (No Response)

Reviewer #2: (No Response)

2. Is the manuscript technically sound, and do the data support the conclusions?

Reviewer #1: Partly

Reviewer #2: Partly

3. Has the statistical analysis been performed appropriately and rigorously? 

Reviewer #1: I Don't Know

Reviewer #2: N/A

4. Have the authors made all data underlying the findings in their manuscript fully available?

Reviewer #1: No

Reviewer #2: Yes

5. Is the manuscript presented in an intelligible fashion and written in standard English?

Reviewer #1: Yes

Reviewer #2: Yes

6. Review Comments to the Author

Reviewer #1: I carefully read the manuscript and the corresponding responses to the raised concerns to both reviewers and noted (at least for my section) that these points are not sufficiently addressed as well as important raw data sets are still missing.

I therefore do not support direct publication at Plos one based on the following concerns.

Concern 1: The LFIA validation in figure 1, as discussed in the first revision, has not been satisfactory addressed. In general, the “Pickering study” validated +/- 50 samples less. Therefore, the reasoning of a limited dataset in the “Pickering study” or an “extended” data set in this study is not valid in this case.

Furthermore, requested raw data (ELISA data, LFIA pictures, neutralization assay setup with appropriate controls) are missing in the .

Concern 2: Responses to concern 2/3 has only been partially addressed. Even though the manuscript includes “a bigger dataset” the current test validation strategy (samples 14+ POS) do still not reflect the “real-life” test-sensitivity from my point of view. The authors claim that they consider this limitation before agreeing to perform the test. Can you include potential POS of these patients in the manuscript, to get a better feeling how to interpret these negative LFIA tests (+/- 60%) in figure 2?

Concern 3: Response to concern 4 has only been partially addressed since comments on the AB quantification and AB dynamics has not been discussed in the manuscript. AB dynamics might represent an important and useful service implementation when included in the service (not only for borderline results) and should be included in the manuscript.

Concern 4: Concern 5 has only been partially addressed in the discussion.

Concern 5: Concern 6 has not been addressed since the Specificity (95% CI) describes a statistical test and the data are at least linked/interpreted like this (Title of the table: Specificity of SureScreen LFIA). The authors should tone down their statements or adjust the table.

Concern 6: Concern 7 has not been addressed. Even though ID50 values were now reported in the results neither an experimental layout with appropriate controls nor plaque pictures or raw data from these assays are included in the manuscript. Implementation of this experiment is necessary to interpret the results and will strengthen the manuscript.

Concern 7: Minor concern 4 is partially addressed. It is hard to believe how those bands can be grouped by eye in the corresponding categories. Moreover, I have not seen any pictures of those bands yet in the manuscript. Can you please provide those LFIA images? Quantification of these pictures would be helpful to categorize those samples into the corresponding group.

Concern 8: Minor concern 4 is not addressed. See above

Concern 9: Minor concern 6 has not been addressed. If there is one potential niche for LFIA tests then at locations with limited lab material. Capillary blood instead of serum preparation could dramatically simplify the test procedure. Therefore, including those Blood/Plasma validations on the LFIA would indeed strengthen the manuscript.

Reviewer #2: The following questions have not been addressed properly.

Q2. It is not visible in the Results section, that these 168 Samples have been randomly selected.

Q8. the authors still mention 7 out of 33 PCR negative patients, but describe only 6 patients. This point has not been clarified.

Q11. this question has not been addressed in the revised manuscript. The changes are not visible.

7. PLOS authors have the option to publish the peer review history of their article (what does this mean?). If published, this will include your full peer review and any attached files.

Reviewer #1: No

Reviewer #2: No

---

## [Author Response · Author response to Decision Letter 1]

19 Mar 2021

Please find included a response to reviewers in our uploaded documents as part of this re-submission.

---

## [Editor Report · Decision Letter 2]

25 Mar 2021

Clinical utility of targeted SARS-CoV-2 serology testing to aid the diagnosis and management of suspected missed, late or post-COVID-19 infection syndromes: results from a pilot service implemented during the first pandemic wave

PONE-D-20-30782R2

Dear Dr. Merrick,

We’re pleased to inform you that your manuscript has been judged scientifically suitable for publication and will be formally accepted for publication once it meets all outstanding technical requirements.

Kind regards,

Michael Nagler, M.D., Ph.D., MSc

Academic Editor

PLOS ONE
---

## [Editor Report · Acceptance letter]

29 Mar 2021

PONE-D-20-30782R2 

Clinical utility of targeted SARS-CoV-2 serology testing to aid the diagnosis and management of suspected missed, late or post-COVID-19 infection syndromes: results from a pilot service implemented during the first pandemic wave 

Dear Dr. Merrick:

I'm pleased to inform you that your manuscript has been deemed suitable for publication in PLOS ONE. Congratulations! Your manuscript is now with our production department. 

Kind regards, 

on behalf of

Prof. Dr. Michael Nagler 

Academic Editor

PLOS ONE